# Explicit and implicit attitudes toward smoking: Dissociation of attitudes and different characteristics for an implicit attitude in smokers and nonsmokers

Xinyue Gao[1], Daisuke Sawamura[2], Ryuji Saito[1], Yui Murakami[2,3], Rika Yano[2], Satoshi Sakuraba[4], Susumu Yoshida[4], Shinya Sakai[2], Kazuki Yoshida[2]*

1 Graduate School of Health Sciences, Hokkaido University, Sapporo, Japan, 2 Faculty of Health Sciences, Hokkaido University, Sapporo, Japan, 3 Faculty of Human Sciences, Hokkaido Bunkyo University, Eniwa, Japan, 4 School of Rehabilitation Science, Health Sciences University of Hokkaido, Ishikari, Japan

* ot-k-yoshida@huhp.hokudai.ac.jp

**Data Availability Statement:** All relevant data are within the paper and its Supporting information files.

## Abstract

Smoking is a global health risk for premature death and disease. Recently, addictive behaviors, like smoking, were considered to be guided by explicit and implicit processes. The existence of a dissociation between the two attitudes in nonsmokers and the causes of the differences in implicit attitudes toward smoking have not been fully investigated. We investigated the explicit and implicit attitudes toward smoking via a self-reported scale and the single category implicit association test (SC-IAT), respectively, among undergraduate and graduate health sciences students. In addition, we applied the drift-diffusion model (DDM) on the SC-IAT and examined the behavioral characteristics that caused differences in implicit attitude toward smoking between smokers and nonsmokers. The results showed the existence of a dissociation between explicit and implicit attitudes toward smoking among nonsmokers. In addition, nonsmokers had a higher decision threshold than smokers and a higher drift rate in the condition where negative words were associated with smoking. Nonsmokers engaged in SC-IAT with more cautious attitudes and responded more easily in a negative condition since it was consistent with their true attitudes. Conversely, smokers did not show a significant difference in the drift rate between the conditions. These results suggested that the differences in an implicit attitude between smokers and nonsmokers were caused by differences in evidence accumulation speed between the positive and negative conditions. The existence of dissociation between implicit and explicit attitudes toward smoking may indicate the difficulty of measuring true attitude in nonsmokers in a questionnaire survey. Additionally, the DDM results explained the difference of implicit attitude between smokers and nonsmokers; it may provide information on the mechanisms of addictive behaviors and a basis for therapy. However, whether these results are affected by cultural differences requires further investigation.

**Funding:** The author(s) received no specific funding for this work.

**Competing interests:** The authors have declared that no competing interests exist.

## Introduction

Smoking is a global health risk for premature death and disease, with more than 8 million people dying annually from smoking-related disease in the world [1]. The health hazards occur both in smokers and those around them. Exposure to secondhand cigarette smoke has been scientifically recognized as a carcinogenic risk [2]. Additionally, a causal relationship has been established with lung cancer, ischemic heart disease, stroke, and respiratory diseases in exposed children [2,3].

Despite the obvious health effects of exposure to environmental cigarette smoke, it was reported that Japanese nonsmokers were unable to explicitly refuse it. Approximately 85% of Japanese students exposed to environmental cigarette smoke felt uncomfortable. However, 95% put up with it without complaining [4]. It has also been reported that although many non-smokers are uncomfortable with those smoking around them, they do not voice their discomfort [5]. Therefore, there can be a discrepancy between the explicit and implicit attitudes toward smoking among Japanese nonsmokers. However, there are no detailed reports on whether a discrepancy between explicit and implicit attitudes towards smoking exists among nonsmokers. Especially, Japan is a collectivistic society, which attaches more importance to groups and society values [6,7]. Thus, if smoking behavior itself is considered less society deviant, they might show tolerant attitude toward smoking. The existence of dissociation between the two attitudes makes it difficult to measure true attitude using questionnaire-based surveys, which may underestimate nonsmokers' negative attitude toward smoking and influence policy making and implementation of public health measures.

Recently, researchers examined health behavior from a dual process perspective [8,9], where an addictive behavior, like drinking and smoking, was guided by two independent systems, explicit and implicit processes. In other words, the effects of implicit and explicit attitudes and/or motivation toward addictive behaviors differ depending on the stage and state of the addictive behaviors [8,9]. Recent studies especially focused on implicit process as it measures the fast, parallel, effortless, and uncontrolled processes [10,11]. Importantly, many studies revealed that smokers' implicit attitude toward smoking was less negative than that of nonsmokers [12–14]. Additionally, smokers who showed higher nicotine dependence had a less negative attitude towards smoking [15,16]. Moreover, a smoking cessation intervention caused negative changes in implicit attitude (i.e., became a negative attitude toward smoking) toward smoking. It was also associated with positive changes in smoking behavior, independent of explicit motivation [17]. Therefore, the implicit attitudes were considered potentially important intervention outcomes.

The most widely used measure of implicit attitude is the Implicit Association Test (IAT) [10,18,19]. The IAT uses the strength of an association between the target and attributes, which the subjects have intrinsically. On task conditions that consisted of subjects' beliefs, their responses were easier and quicker than on conditions that were inconsistent with their beliefs, which resulted in slower responses [19]. This difference was used to measure the implicit beliefs that the subjects potentially held.

The IAT has been used in various addiction studies, including smoking [13–17]. However, the cause of differences and changes in implicit attitude towards smoking in smokers and non-smokers is unclear. The drift-diffusion model (DDM) is often used in two alternative force choice tasks, like the IAT. The DDM, a type of sequential sampling model, assumes that a choice is a process comprising a noisy accumulation of evidence from a stimulus [20,21]. Several parameters can be estimated, such as drift rate, decision threshold, and non-decision time from the distributions of choice probabilities and reaction time. Additionally, these parameters provide a deeper insight into choice features among the subjects [22,23]. Implicit attitude is an

important surrogate indicator of smoking behavior and can be an intervention outcome [17]. Improving its understanding may provide important suggestions for overcoming addictions.

This study examined the existence of a dissociation between the explicit and implicit attitudes toward smoking among nonsmokers and the behavioral characteristic differences that caused differences in implicit attitudes toward smoking between smokers and nonsmokers. We investigated the explicit and implicit attitudes toward smoking using a self-reported scale and the single category implicit association test (SC-IAT) [24], respectively, in undergraduate and graduate students. Moreover, we applied the DDM to the SC-IAT data to explore the behavioral characteristics that caused differences in implicit attitude toward smoking. We hypothesized that a) the correlation between explicit and implicit attitudes would not be significant among nonsmokers due to the dissociation of these attitudes and b) the drift rate, the speed of evidence accumulation, would be higher among nonsmokers than smokers.

## Materials and methods

### Sample size estimation

The required sample size was determined by a priori power analysis using G*Power 3.1.1 [25]. Ren et al. [26] reported the effect size between two groups in D-score from 0.886 to 1.277. We adopted a lower effect size as a conservative estimate. The sample size was estimated based on significance probability ($\alpha = .05$), statistical power ($1-\beta = .80$), and effect size (Cohen's d = 0.886), which resulted in 18 in one group. Considering a 15–20% dropout and outlier rate, we set 24 in one group as desirable.

### Subjects

We recruited 48 healthy subjects (Table 1). To ensure homogeneity, we recruited both under-graduate and graduate students aged 20–30 years from the Department of Health Sciences in Hokkaido and Hokkaido Bunkyo Universities. Current smokers and nonsmokers were those who smoked at least weekly and had never smoked before or had smoked less than 10 cigarettes till date, respectively [26]. Those who had currently stopped smoking were excluded.

**Table 1. Participants' characteristics.**

|  | Non-smoker | Smoker | p value |
|---|---|---|---|
| n | 24 | 24 |  |
| Age [a] | 22.96 ± 1.73 | 21.50 ± 0.83 | < .001 |
| Gender [b] | Male: 18<br>Female: 6 | Male: 21<br>Female: 3 | .46 |
| Are there any smoker around you? [b] | Yes: 13<br>No: 11 | Yes: 24<br>No: 0 | < .001 |
| FTND |  | 2.21 ± 2.28<br>Range: 0–7 |  |
| Number of cigarettes smoked per day |  | 8.50 ± 7.13<br>Range: 1–30 |  |
| Duration of smoking (months) |  | 16.15 ± 10.28<br>Range: 2–54 |  |

Note: Values are mean ± s.d. or n.

[a]Unpaired t-test.

[b]Fisher's exact test. FTND: Fagerstrom Test for Nicotine Dependence.

The Ethics Committee of the Faculty of Health Sciences at Hokkaido University approved the study protocol (approval number 20-42-3). All participants provided written informed consent.

## Explicit attitude for smoking

The Kano Test for Social Nicotine Dependence (KTSND) [27] was used to measure explicit attitudes for smoking. It consisted of 10 questions regarding smoking rated (e.g., Smokers' lifestyles may be respected, Smoking sometimes enriches people's life) on a scale from 0 (disagree) to 3 (agree). A score of > 10 indicated a strong tendency to accept smoking and deny its harmfulness. A higher score indicated a higher attitude to accept, affirm, and tolerate smoking. This scale can be used for both smokers and nonsmokers, and its reliability (Cronbach's α = 0.77) and validity were confirmed in a past study [28].

## Implicit attitude for smoking

The Single Category Implicit Association Test (SC-IAT) was used to measure subjects' implicit attitudes toward smoking (Fig 1). It included eight positive words (joy, happy, gentle), eight negative words (dirty, annoying, incompetent), and eight smoking images (a person smoking, lit cigarette, and a lighter in the box). All the pictures were collected from the Internet (Google image search) with reference to a previous research (17). Furthermore, the positive and negative words were selected from a pool of words from previous studies [13,26,29,30] that reflected attitudes toward smoking. The SC-IAT consisted of four blocks. Blocks 1 & 3 had 24 trials each and blocks 2 & 4 had 72 trials each. In blocks 1 & 2 (positive condition: smoking images were set with positive words), two labels, positive and smoking, were presented at the top left corner of the display, while the label, negative, was presented at the top right corner. In blocks 3 & 4 (negative condition: smoking images were set with negative words), two labels, negative and smoking, were presented at the top right corner while the label, positive, was presented at the top left corner. Subjects were asked to categorize each image or word presented in the center by pressing "E" (left) or "I" (right) key on the keyboard. If the subjects categorized the target image or word to the incorrect side, a red "X" appeared at the center and they had to press the other (correct) key. No feedback was presented after the correct response, and the inter-trial interval was 0.5 s. Referring to [24,31] studies, in blocks 1 & 2, smoking images, positive words, and negative words were presented in a 7:7:10 ratio. Similarly, in blocks 3 & 4, they were presented in a 7:10:7 ratio. The performance conditions order was counterbalanced across subjects. This task was made and presented by PsychoPy (v3.1.0) [32].

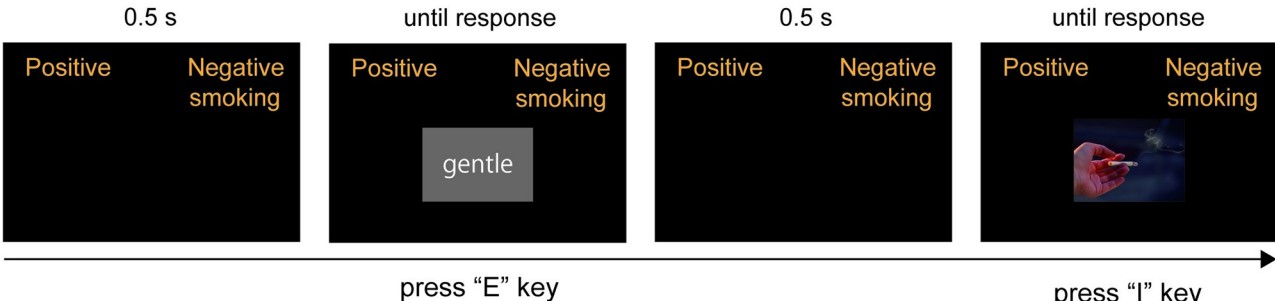

**Fig 1. Schematic figure of the single category implicit association test (SC-IAT).** Interstimulus interval was set at 0.5 s and stimuli were presented until subjects' response. If subjects make an error, a red "X" is presented in the center, and they are required to correct their answer. All words were presented in Japanese.

The D-score of the SC-IAT was calculated by the following procedure with reference to [19]. First, trials whose reaction time (RT) was greater than 10,000 ms were excluded, and the mean RT for all four blocks was computed. Second, we computed individual pooled standard deviations (SD) for blocks 1 & 3 and blocks 2 & 4. Third, we calculated the two differences: 1) mean RT in Block 3 –mean RT in Block 1 and 2) mean RT in Block 4 –mean RT in Block 2. Fourth, each difference value was divided by the corresponding pooled SD, calculated in step 2. Finally, we calculated the mean of the two results obtained in step 4, which was the individual's SC-IAT D score. This calculation was performed only for the correct trials.

## Procedure

To avoid a large variance in scale responses due to fluctuation in nicotine craving, the smokers were asked to not smoke two hours before the participation [33]. After informed consent was obtained, all the subjects were asked to complete a questionnaire to collect basic information, such as age, sex, and presence of smokers near them. For smokers, they answered regarding their smoking habit (e.g., "how long have you been smoking?" and "how many cigarettes do you smoke per day on average?"). Additionally, they also answered the Fagerstrom Test for Nicotine Dependence (FTND) [34] which assessed the severity of nicotine dependence. The FTND, widely used to assess nicotine dependence (e.g., How soon after you wake up do you smoke your first cigarette?), has enough reliability (Cronbach's $\alpha = 0.70$) and validity [35]. It consisted of six items, and a higher score indicated more severe nicotine dependence. Next, all subjects completed the SC-IAT. After they were given instruction on how to implement it, 24 practice trials that only used words were conducted to familiarize them. The practice trials used different words from those used in the main task and did not use any images. Lastly, they completed the KTSND.

## Drift-Diffusion Model (DDM)

To explore the cause of difference in explicit and implicit attitudes toward smoking in both groups, we employed the DDM. We collected the subjects' first response RT and accuracy data in each trial to use the DDM for the SC-IAT. We conducted the hierarchical Bayesian estimation of the DDM parameters for each subject using the HDDM toolbox [36] in python. To ensure independency of the estimated parameter, each subject's data was fit separately and not incorporated into the hierarchical model. Our model had three free parameters: non-decision time ($t$), decision threshold ($a$), and drift rate ($v$). Further, the starting point was fixed at $a/2$. The upper and lower boundaries indicated correct and incorrect responses, respectively. The HDDM used Markov-chain Monte Carlo (MCMC) sampling to approximate the posterior distribution over parameter estimates. For parameter estimation, three chains were run each with 2000 samples and the first 500 samples in each were discarded as burn in. Furthermore, Gelman and Rubin's R̂ for each parameter was calculated to assess convergence. Mean posterior estimates parameters were extracted for the subsequent statistical tests.

## Statistical analysis

Demographic data were analyzed using unpaired t-test and Fisher's exact test. The differences of explicit and implicit attitudes for smoking between smokers and nonsmokers were evaluated using unpaired t-test. To examine whether explicit and implicit attitudes for smoking were consistent within each group, Pearson's correlation analysis was conducted between the KTSND and D-scores. For smokers, we conducted an additional correlation analysis to examine the FTND score and whether the number of cigarettes smoked per day correlated with explicit and implicit attitudes toward smoking. Each parameter ($a$, $t$, $v$) estimated by the DDM

was analyzed by a two-way analysis of variance (ANOVA) with the group (smokers or non-smokers) and condition (positive or negative) as factors. If an interaction was found, we conducted a post-hoc analysis. All analyses were performed with R 4.1.0 and R studio (https://www.rstudio.com/), and the level of significance was set at $p < .05$.

## Results

### Demographic data

Table 1 shows the participants' demographic data. There was a significant difference in age (t (46) = 3.71, $p < .001$, 95% confidence interval [CI] [0.69–2.25]) and the presence of smokers in the surroundings (odds ratio [OR] = 0, $p < .001$, 95%CI [0.00–0.27]) between nonsmokers and smokers. However, there was no significant difference in gender (OR = 0.44, $p = .46$, 95% CI [0.06–2.4]).

### Explicit and implicit attitudes for smoking and the correlations

First, we investigated whether there was a group difference between explicit and implicit attitudes toward smoking. The KTSND scores and SC-IAT D scores, which represented explicit and implicit attitudes for smoking, respectively, were significantly higher in smokers than nonsmoker (t(46) = -3.61, $p < .001$, 95%CI [-5.91–1.68]); t(46) = -2.21, $p = .03$, 95%CI [-0.48–0.18]) (Fig 2a and 2b). Next, we investigated whether each score was significantly higher or lower than the criterion score (9 for KTSND and 0 for SC-IAT D-score) via a one-sample t-test. In the KTSND, both groups showed significantly higher scores than the criterion score (smokers: t(23) = 11.79, $p < .001$; nonsmoker: t(23) = 5.24, $p < .001$). Furthermore, in the SC-IAT D-score, the nonsmokers showed significantly lower scores than the criterion score (t (23) = -6.38, $p < .001$). However, smokers did not show a significant difference (t(23) = -1.60, $p = .062$). Next, we investigated whether an explicit attitude for smoking correlated with the implicit attitude in each group. There was a significantly positive correlation between the

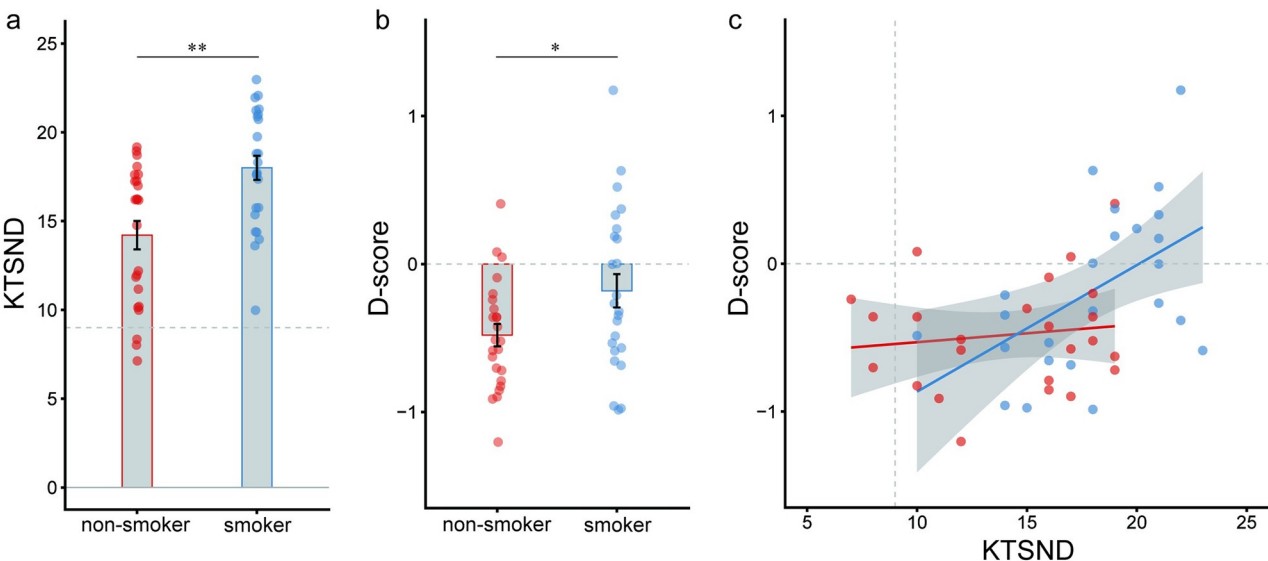

**Fig 2. The right panel depicts the group difference of the KTSND score, the center panel depicts the group difference of SC-IAT D-score, and the left panel depicts the difference of correlation between KTSND and SC-IAT D-scores.** Nonsmokers data is shown in orange and smokers' in blue. Error bars indicates a standard error and grey dashed line indicates the boundary that distinguishes between the positive and negative attitude toward smoking. KTSND: Kano Test for Social Nicotine Dependence. *p < .05, **p < .01.

KTSND and SC-IAT D-scores in smokers (r = 0.514, $p$ = .01); however, not in nonsmokers (r = 0.128, $p$ = .55) (Fig 2c). Additionally, to formally evaluate whether the correlation between explicit and implicit attitudes for smoking was different between the two groups, the interaction effect of a linear model (SC-IAT D-score ~ KTSND*group) was evaluated. We found a significant effect of the KTSND (β = 0.049, 95%CI [0.014–0.083], $p$ < .01) and a significant interaction of KTSND*group (β = -0.037, 95%CI [-0.072–0.002], $p$ < .05). These results indicated that the effect of the KTSND on SC-IAT D-score was significantly different by group, which suggested that the degree of congruence between explicit and implicit attitudes toward smoking differed between smokers and nonsmokers.

### Correlation analysis in the smoker group

To investigate the association between smoking status (FTND and the number of cigarettes smoked per day) and both attitudes for smoking, a correlation analysis was conducted. There were medium to high positive correlations between FTND and KTSND scores and the number of cigarettes smoked (r = 0.8, 95%CI [0.581–0.908], $q$ < .001; r = 0.52, 95%CI [0.142–0.760], $q$ < .05), respectively. Furthermore, an association between the KTSND and FTND scores (r = 0.45, 95%CI [0.060–0.724], $q$ < .05) was found. These analyses were controlled for the false discovery rate (FDR) (Fig 3).

### Drift-diffusion model for implicit attitude for smoking

Fig 4 shows the results of the two-way ANOVA with the group and condition as factors in DDM analysis. We found a significant main effect of group in decision threshold (F(1,92) = 4.50, $p$ = .036, $\eta^2$ = 0.05) and non-decision time (F(1,92) = 4.65, $p$ = .033, $\eta^2$ = 0.05), and significant main effect of condition in drift rate (F(1,92) = 9.89, $p$ < .01, $\eta^2$ = 0.09). Although we did not find a significant interaction effect in drift rate (F(1, 92) = 2.35, $p$ = .128, $\eta^2$ = 0.02), we conducted a paired t-test to examine the differences in response between the two conditions in each group. The drift rate in the negative condition was significantly higher than in the positive condition in nonsmokers (t(23) = 3.97, $p$ < .001). However, there was no significant drift rate difference between the conditions in smokers (t(23) = 1.31, $p$ = .202). The ranges of the R̂ value in all parameter estimates indicated satisfactory convergence (a: 0.999–1.025, t: 0.999–1.019, and v: 0.999–1.019).

### Discussion

Smokers consistently showed more positive and permissive attitudes toward smoking than nonsmokers. Interestingly, nonsmokers' explicit and implicit attitudes toward smoking were dissociated, and they showed an ostensibly positive attitude even though they were viscerally negative toward smoking. To investigate the causes of the differences between smokers' and nonsmokers' implicit attitudes toward smoking, the DDM was applied to the SC-IAT. From the estimated parameters, we found that nonsmokers behaved more cautiously in the SC-IAT than smokers and responded more easily in the negative condition than the positive condition.

We found a significant positive correlation between KTSND and SC-IAT D-scores in smokers; however, not in nonsmokers. Additionally, there was a significant interaction effect between KTSND and group on SC-IAT D-scores. These results revealed that the smokers' both attitudes toward smoking were relatively consistent; however, those of nonsmokers were not. Particularly, nonsmokers showed positive and permissive attitudes and negative attitudes in their explicit and implicit attitudes, respectively. The result may have been influenced by Japanese culture where collective society is respected. Japanese communication styles were indirect verbal expression and implication embedded in nonverbal communication, and they

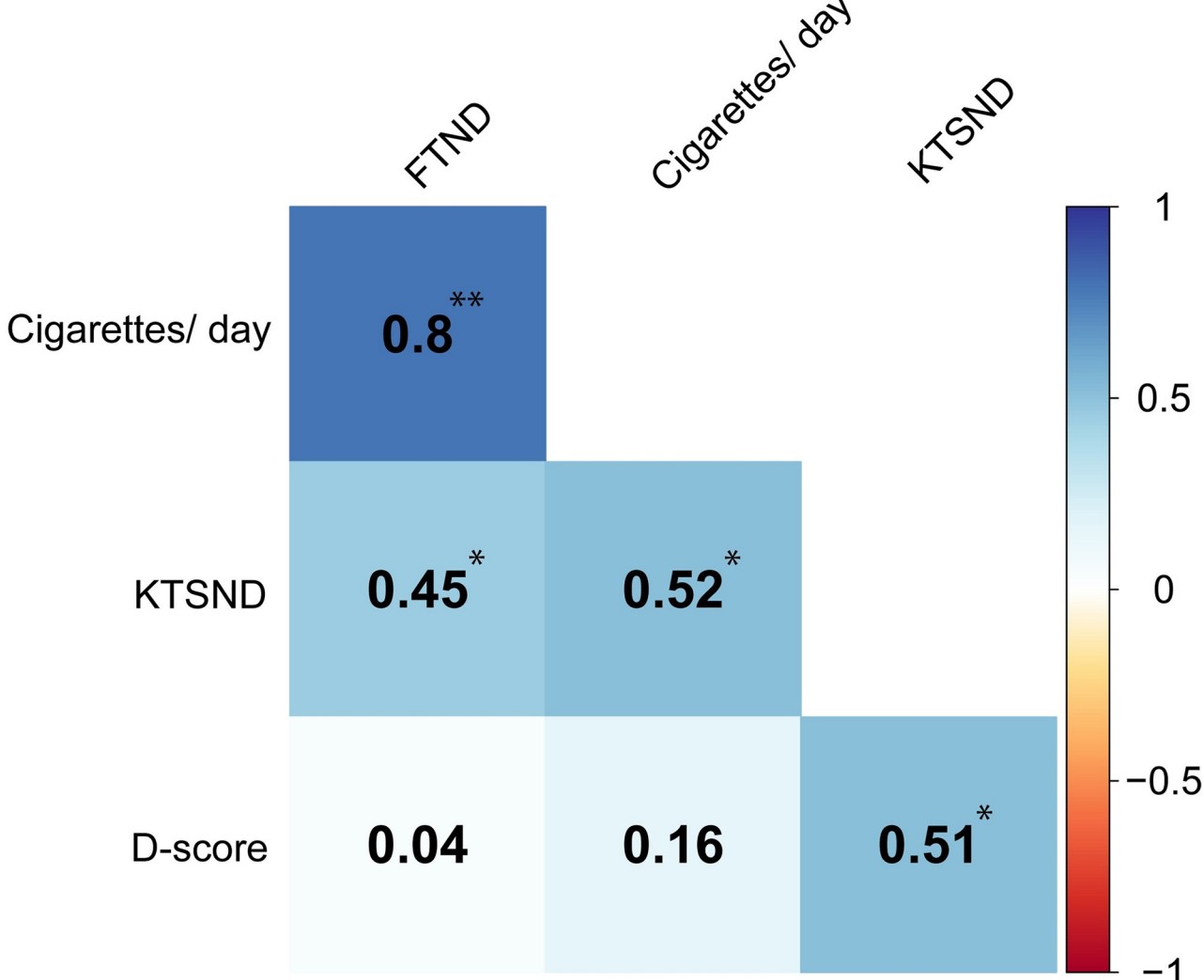

**Fig 3. Correlation analysis among the smoking-related measures.** All p-values were corrected with FDR correction. FTND: Fagerstrom Test for Nicotine Dependence, KTSND: Kano Test for Social Nicotine Dependence. $^*q < .05$, $^{**}q < .01$.

value group harmony [6,7]. In these cultures, it was possible that nonsmokers considered it more socially desirable to have an explicitly positive attitude toward smoking rather than disturbing the collective society. Social desirability bias is often referred to in studies of prejudice and deception as a cause of dissociation between explicit and implicit attitudes [37,38]. Indeed, a previous study on students in the Netherlands reported that both attitudes toward smoking were consistent even among nonsmokers [15]. Although there were some methodological differences (e.g., semantic differential was used as explicit measure in the previous study), this different result may suggest cultural differences in explicit and implicit attitudes toward smoking.

We found a significant positive correlation between FTND and the number of cigarettes smoked per day among smoking status. Among the ingredients in cigarettes, nicotine was closely associated with tobacco dependence [39] and had an effect on the mesolimbic dopaminergic reward system [40]. It was associated with both increased pleasure and reduced negative effects of smoking [41]. Considering these nicotine dependence mechanisms, the significant positive correlation between FTND and the number of cigarettes smoked per day is plausible.

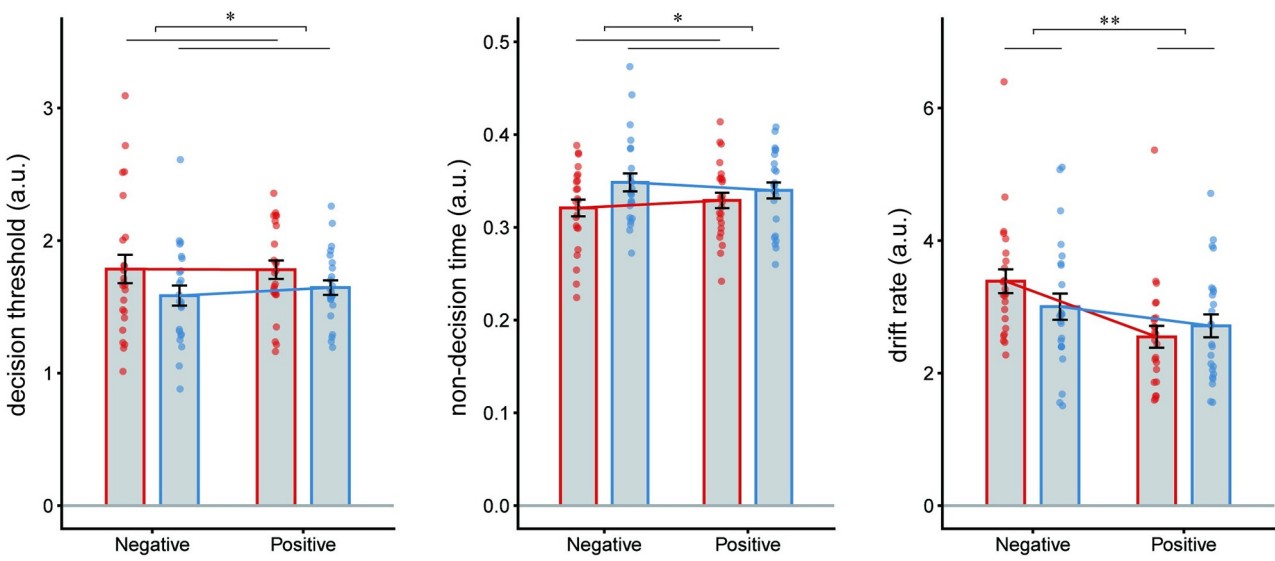

**Fig 4. The ANOVA results estimated DDM parameters decision threshold (left panel), non-decision time (center panel), and drift rate (right panel).** The results show a significant main effect of group in decision threshold and non-decision time, and a significant main effect of condition in drift rate. Nonsmokers' data is shown in orange and smokers' in blue. Error bars indicates standard error. $^*p < .05$, $^{**}p < .01$.

Furthermore, we also found a significant positive correlation between the KTSND and FTND and number of cigarettes smoked per day. However, the SC-IAT D-scores were not correlated with any smoking status indices. Swanson et al [14] reported that smokers bolstered their explicit attitudes toward smoking. Hence, these explicit attitudes were consistent with their smoking behavior. From this report and our results, smokers bolster explicit attitudes consistent with their own smoking behavior could be the reason for the moderate correlation between smoking status and explicit attitudes toward smoking.

From the DDM analysis of the SC-IAT data, nonsmokers showed significantly higher decision threshold and lower non-decision time than smokers and significantly higher drift rate in negative task condition than positive task condition. These results may indicate that nonsmokers had more cautious attitudes for responding to smoking-related stimuli than smokers and faster evidence accumulation in the negative task condition. Since the higher drift rate indicated that the decision was easier [42,43], it can be inferred that the negative condition was easier for nonsmokers. It is plausible to interpret that nonsmokers had a strong association between cigarettes and negative words, consistent with their beliefs, and therefore, responded easier to the SC-IAT. Additionally, a higher decision threshold, which indicated a cautious attitude for SC-IAT task, could reflect a cautious attitude for engaging in smoking-related tasks. Conversely, smokers did not show significant differences in the drift rate between the task conditions, suggesting that there were no differences between the conditions to bias their response in task difficulties. Hence, they may have a relatively neutral belief regarding smoking. The nondecision time in smokers was significantly higher than in nonsmokers. This result indicated that smokers required more time for motor response and to encode the task. It was generally reported that reaction time was longer with aging, and was expressed by an increased decision threshold and non-decision time [21]. However, in our study, the age difference between the two groups was not considered a clinically meaningful difference in the context of aging. Further, there was a lack of enough evidence to suggest that this was an effect of smoking. An advantage of applying the DDM was that many studies have examined the correspondence between DDM's each parameter and neural substrate. It has been reported that the

prefrontal cortex, frontal eye field, lateral intraparietal area, and medial prefrontal cortex effected drift rate parameter, and the subthalamic nucleus and pre-supplementary motor area effected decision threshold [44–47]. Recently, noninvasive brain stimulation on the dorsolateral prefrontal cortex has been reported to be effective as a smoking cessation treatment [48], and this stimulation site coincides with an area that is related to drift rate. Hence, the regions of the brain associated with DDM parameters for implicit attitudes toward addictive behaviors should be examined. This may provide information on the mechanisms of addictive behaviors and a basis for non-invasive brain stimulation therapy [48].

We must note several limitations. First, we included only young and health science course students to ensure homogeneity. These students were more health conscious and knowledgeable than the general population, which may have influenced the results of attitudes toward smoking. Second, the applicability of the results to different cultural background is limited. Our results might be influenced by the Japanese cultural context such as collectiveness society; therefore, future studies should examine the applicability of these results in another cultural context. Third, our sample was relatively small. Although the calculated sample size was adhered to and examined for hypothesis testing, confirming reproducibility with a larger, more general population would be useful to expand the generalizability of this study.

In conclusion, we showed the existence of dissociation between explicit and implicit attitude toward smoking in nonsmokers, and that the differences in implicit attitude between smokers and nonsmokers were due to differences in the evidence accumulation rate in SC-IAT. These findings suggested that in a culture where collective society is respected, the questionnaire-based survey might not reflect the true attitudes of nonsmokers. Therefore, we must consider the attitude dissociations when making public policy choice for regulation of smoking based on a questionnaire survey. Additionally, DDM results would contribute to the further understanding of mechanisms, neural response, and treatments for addictive behavior like smoking.

## Supporting information

**S1 Table. Data set underlying the results.**
(XLSX)

## Author Contributions

**Conceptualization:** Xinyue Gao, Kazuki Yoshida.

**Data curation:** Xinyue Gao, Ryuji Saito, Kazuki Yoshida.

**Formal analysis:** Xinyue Gao, Kazuki Yoshida.

**Investigation:** Xinyue Gao, Ryuji Saito, Yui Murakami, Rika Yano, Satoshi Sakuraba, Susumu Yoshida.

**Methodology:** Kazuki Yoshida.

**Supervision:** Shinya Sakai.

**Validation:** Xinyue Gao, Daisuke Sawamura, Shinya Sakai, Kazuki Yoshida.

**Visualization:** Kazuki Yoshida.

**Writing – original draft:** Xinyue Gao, Kazuki Yoshida.

**Writing – review & editing:** Daisuke Sawamura, Rika Yano, Shinya Sakai.

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
