## [Decision Letter · Decision Letter 0]

12 Sep 2022

PONE-D-22-17788Explicit and implicit attitudes toward smoking: dissociation of attitudes and different characteristics for an implicit attitude in smokers and nonsmokersPLOS ONE

Dear Dr. Yoshida,

Thank you for submitting your manuscript to PLOS ONE. After careful consideration, we feel that it has merit but does not fully meet PLOS ONE’s publication criteria as it currently stands. Therefore, we invite you to submit a revised version of the manuscript that addresses the points raised during the review process.

We look forward to receiving your revised manuscript.

Kind regards,

Billy Morara Tsima, MD MSc

Academic Editor

PLOS ONE

Journal Requirements:

Additional Editor Comments:

The manuscript contains a number of grammatical errors that need to be corrected to improve readability and provide clarity.

Line 31; rephrase...among undergraduate and graduate health sciences students. (or use "a" instead of "the"...health sciences course.

Line 63; complaining vs complain

Line 70; thus vs hus

Line 79-81; the sentence needs to be reviewed. It is not clear what the pronoun "it" refers to in this context. Does it refer to "recent studies"? in which case "they" would be more appropriate than "it". Or, does it refer to "implicit process"? in which case it would be more appropriate to say "measure" as opposed to "measured" in this context.

Line 84; an "a" is missing before smoking cessation intervention

Line 91-93; Need to clarify what is meant by trials in this context. It is not intuitive that the authors are referring to repeated testing using the tool as this is not explained prior to using the term. Since the strengths of the tool is being discussed, the word trial can imply a clinical trial.

Line 109-111;The sentence is not clear and needs to be rephrased. ("the both attitudes" is not grammatically correct)

Line 114; The sentence is not clear and needs to be rephrased. (would not be significant vs would be not significant)

Line 347; is plausible vs was plausible

Line 357-358; Rephrase for clarity

Line 361; is plausible vs was plausible

Line line 366-368; Needs rephrasing. Saying that "it was suggested" distances the discussion from the present results. The authors should discuss the results presented in the manuscript and therefore use present tense when referring to the results

Line 372-374; This needs to be rephrased. The statement implies that the authors were testing the hypothesis that smoking makes one younger. The foregoing discussion on aging does not appear to be linked to the finding on the age difference between the two groups. Although the difference was statistically significant, the difference does not appear to be clinically meaningful in the context of aging.

Line 380-381; "where" is not appropriately placed

Reviewers' comments:

Reviewer's Responses to Questions

**Comments to the Author**

1. Is the manuscript technically sound, and do the data support the conclusions?

Reviewer #1: No

Reviewer #2: Yes

2. Has the statistical analysis been performed appropriately and rigorously? 

Reviewer #1: Yes

Reviewer #2: Yes

3. Have the authors made all data underlying the findings in their manuscript fully available?

Reviewer #1: Yes

Reviewer #2: No

4. Is the manuscript presented in an intelligible fashion and written in standard English?

Reviewer #1: Yes

Reviewer #2: Yes

5. Review Comments to the Author

Reviewer #1: This is an excellent article which summarize the main topics of the study, arguments, positions, analysis and findings. In my point of view, the paper provides new knowledge of this field of research and it can be published with no additional revisions

Reviewer #2: I appreciate the opportunity given to review this interesting and informative paper on explicit and implicit attitudes toward smoking: Dissociation of attitudes and different characteristics for an implicit attitude in smokers and nonsmokers.

The focus of the study is clearly defined in terms of what aspects of attitudes toward smoking are under observation (implicit and explicit attitudes). The originality of the research is unquestionable as it differs from other literature on the differences in attitudes toward smoking between smokers and nonsmokers since it emphasizes the role of both implicit and explicit attitudes. The study also shows adequate knowledge and understanding of relevant literature on the concept of attitudes toward smoking.

Variables used in the study are well defined. Data analysis was properly done and results were well presented. The results from the sample used in the study presented adds to the growing literature on the concept.

The discussion section is well written. The authors make adequate references to other studies that relate to the outcome of their study. Also, the sentences are clearly expressed and readable.

However, there were a few grammatical errors identified. For instance, the authors wrote “hus” instead of “Thus” in line 70. It is recommended that the authors would revise the manuscript to check for grammatical errors.

Again I am grateful for the opportunity to review this manuscript and hope to see it in print.

6. PLOS authors have the option to publish the peer review history of their article (what does this mean?). If published, this will include your full peer review and any attached files.

Reviewer #1: No

Reviewer #2: No

---

## [Author Response · Author response to Decision Letter 0]

19 Sep 2022

Response to reviewers

[Reviewer #1]

This is an excellent article which summarize the main topics of the study, arguments, positions, analysis and findings. In my point of view, the paper provides new knowledge of this field of research and it can be published with no additional revisions.

Response:

We thank the reviewer for carefully reading our manuscript and providing positive comments.

[Reviewer #2]

I appreciate the opportunity given to review this interesting and informative paper on explicit and implicit attitudes toward smoking: Dissociation of attitudes and different characteristics for an implicit attitude in smokers and nonsmokers.

The focus of the study is clearly defined in terms of what aspects of attitudes toward smoking are under observation (implicit and explicit attitudes). The originality of the research is unquestionable as it differs from other literature on the differences in attitudes toward smoking between smokers and nonsmokers since it emphasizes the role of both implicit and explicit attitudes. The study also shows adequate knowledge and understanding of relevant literature on the concept of attitudes toward smoking.

Variables used in the study are well defined. Data analysis was properly done and results were well presented. The results from the sample used in the study presented adds to the growing literature on the concept.

The discussion section is well written. The authors make adequate references to other studies that relate to the outcome of their study. Also, the sentences are clearly expressed and readable.

However, there were a few grammatical errors identified. For instance, the authors wrote “hus” instead of “Thus” in line 70. It is recommended that the authors would revise the manuscript to check for grammatical errors.

Again I am grateful for the opportunity to review this manuscript and hope to see it in print.

Response:

We thank the reviewer for carefully reading our manuscript and providing positive comments. Following the suggestions of Reviewer #2 and the Academic Editor, we have revised the manuscript and checked for grammatical errors. The changes are shown in red font in the main manuscript.

[Journal Requirements and Editor comments]

We thank the editor for carefully reading our manuscript and identifying grammatical errors. We revised the manuscript and asked Editage to proofread it. The changes are shown in red font in the main manuscript.

[Response]

We revised our manuscript to meet PLOS ONE’s style requirements. 

[Response]

We have provided the minimal data set underlying the results described in our manuscript as supporting information files. Please confirm it. 

3. Please review your reference list to ensure that it is complete and correct.

[Response]

We have checked our reference list. We removed reference No.23 and replaced it relevant reference. The changes are shown in red font in the main manuscript.

Additional Editor Comments:

[Response]

We revised the manuscript with respect to all of the following points and asked Editage to proofread it. The changes are shown in red font in the main manuscript. The line numbers of the revised text are indicated below in bold text.

Line 31; rephrase...among undergraduate and graduate health sciences students. (or use "a" instead of "the"...health sciences course. (Lines 31–32) 

Line 63; complaining vs complain (Line 67)

Line 70; thus vs hus (Line 74)

Line 79-81; the sentence needs to be reviewed. It is not clear what the pronoun "it" refers to in this context. Does it refer to "recent studies"? in which case "they" would be more appropriate than "it". Or, does it refer to "implicit process"? in which case it would be more appropriate to say "measure" as opposed to "measured" in this context. (Line 84)

Line 84; an "a" is missing before smoking cessation intervention (Line 88)

Line 91-93; Need to clarify what is meant by trials in this context. It is not intuitive that the authors are referring to repeated testing using the tool as this is not explained prior to using the term. Since the strengths of the tool is being discussed, the word trial can imply a clinical trial. (Line 95–96)

Line 109-111;The sentence is not clear and needs to be rephrased. ("the both attitudes" is not grammatically correct) (Line 113)

Line 114; The sentence is not clear and needs to be rephrased. (would not be significant vs would be not significant) (Line 118)

Line 347; is plausible vs was plausible (Line 353)

Line 357-358; Rephrase for clarity (Line 363–365)

Line 361; is plausible vs was plausible (Line 367)

Line line 366-368; Needs rephrasing. Saying that "it was suggested" distances the discussion from the present results. The authors should discuss the results presented in the manuscript and therefore use present tense when referring to the results (Line 372)

Line 372-374; This needs to be rephrased. The statement implies that the authors were testing the hypothesis that smoking makes one younger. The foregoing discussion on aging does not appear to be linked to the finding on the age difference between the two groups. Although the difference was statistically significant, the difference does not appear to be clinically meaningful in the context of aging. (Line 378–381)

Line 380-381; "where" is not appropriately placed (Line 388)

---

## [Editor Report · Decision Letter 1]

26 Sep 2022

Explicit and implicit attitudes toward smoking: dissociation of attitudes and different characteristics for an implicit attitude in smokers and nonsmokers

PONE-D-22-17788R1

Dear Dr. Yoshida,

We’re pleased to inform you that your manuscript has been judged scientifically suitable for publication and will be formally accepted for publication once it meets all outstanding technical requirements.

Kind regards,

Billy Morara Tsima, MD MSc

Academic Editor

PLOS ONE
---

## [Editor Report · Acceptance letter]

29 Sep 2022

PONE-D-22-17788R1 

Explicit and implicit attitudes toward smoking: dissociation of attitudes and different characteristics for an implicit attitude in smokers and nonsmokers 

Dear Dr. Yoshida:

I'm pleased to inform you that your manuscript has been deemed suitable for publication in PLOS ONE. Congratulations! Your manuscript is now with our production department. 

Kind regards, 

on behalf of

Dr. Billy Morara Tsima 

Academic Editor

PLOS ONE